

# Identifying the predisposing factors, signs and symptoms of overreaching and overtraining in physical education professionals

Ricardo B. Viana[1], Paulo Gentil[1], Vinício S. Lorenço[1], Carlos A. Vieira[1], Mário H. Campos[1], Douglas A.T. Santos[1,2], Wellington F. Silva[1], Marilia S. Andrade[3], Rodrigo L. Vancini[4] and Claudio A.B. de Lira[1]

[1] Faculdade de Educação Física e Dança, Universidade Federal de Goiás, Goiânia, Goiás, Brazil
[2] Colegiado de Educação Física, Universidade do Estado da Bahia, Teixeira de Freitas, Bahia, Brazil
[3] Departamento de Fisiologia, Universidade Federal de São Paulo, São Paulo, São Paulo, Brazil
[4] Centro de Educação Física e Desportos, Universidade Federal do Espírito Santo, Vitória, Espírito Santo, Brazil

Corresponding author
Claudio A.B. de Lira,
andre.claudio@gmail.com

## ABSTRACT

**Background**. It is possible that physical education professionals, especially those who participate in aerobic activities, have predisposing factors, signs and symptoms of overreaching (OVR) and overtraining (OVT) due to a high load and volume of exercise followed by suboptimal recovery time. The present study aimed to identify the predisposing factors, signs and symptoms of OVR and OVT in physical education professionals.

**Methods**. A questionnaire consisting of 42 questions (10 questions group) about predisposing factors and signs/symptoms was answered by 132 physical education professionals from both sexes (83 men and 49 women) who were allocated into a resistance training group (RG, $n = 74$), aerobic training group (AG, $n = 20$) and resistance and aerobic training group (RAG, $n = 38$). A mean score was calculated ranging from 1 (completely absent) to 5 (severe) for each question group. A low occurrence of predisposing factors and signs and symptoms of OVR and OVT was considered to be a question group score 4 or lower. Profile of Mood State Questionnaire (POMS) was also applied.

**Results**. A mean score of $2.5 \pm 0.7$, $2.7 \pm 0.7$ and $2.7 \pm 0.8$ was found for all question groups for RG, AG and RAG, respectively. Of the total sample, 40.6% trained at least five times/week. The POMS revealed that 67.5% of the RG ($n = 50$), 80% of the AG ($n = 16$) and 60.5% of the RAG ($n = 23$) were classified as having no mood disorders and a standard graphic iceberg was presented. There were no statistical differences ($p > 0.05$) in the total mood disorders among RG ($13.9 \pm 24.5$), AG ($10.3 \pm 25.1$) and RAG ($14.6 \pm 27.9$) groups.

**Conclusion**. Despite the volume of training/body working performed by the physical education professionals surveyed being greater than the recommended to achieve improvements on physical fitness, they did not show predisposing factors, signs or symptoms of OVR and OVT.

## INTRODUCTION

Overtraining syndrome (OVT) is characterized by multiple signs and symptoms. It involves the accumulation of stress, leading to a decrease in physical/mental/working performance during the sports season (*Budgett, 1990*). OVT is commonly attributed to an imbalance between training/working stimulus (generally, aerobic training (*Elliott, Wagner & Chiu, 2007*)) and rest periods, therefore, the individual is subjected to high intensity and/or high volumes without an appropriate recovery time (*Lemyre, Roberts & Stray-Gundersen, 2007*; *Jones & Tenenbaum, 2009*; *Meeusen et al., 2013*). Psychological and/or social stressors can also contribute to this syndrome (*Kreher & Schwartz, 2012*). Apart from a decrease in performance, OVT syndrome is usually accompanied by physiological, immunological, biochemical and psychological alterations (*Fry, Morton & Keast, 1991*; *Lehmann, Foster & Keul, 1993*; *Kuipers, 1998*; *Carfagno & Hendrix, 2014*). For example, the following alterations are common in OVT: parasympathetic (fatigue, depression, bradycardia and loss of motivation), sympathetic (insomnia, irritability, tachycardia, agitation, hypertension and restlessness) and others (anorexia, weight loss, lack of concentration, anxiety, heavy/stiff muscle, and awakening unrefreshed) (*Carfagno & Hendrix, 2014*).

During some periods of the sports season, such as a shock microcycle, coaches can plan an intentional increase in volume/intensity of training load in order to potentiate physiological adaptations that would result in a chronic increase of performance (*Issurin, 2010*). Frequently, this training manipulation leads the athlete to develop transitory signs and symptoms similar to OVT syndrome known as overreaching (OVR) (*Fry, Morton & Keast, 1991*; *Lehmann, Foster & Keul, 1993*; *Kuipers, 1998*). However, OVR recovery can occur a few days after refraining from training or decreasing exercise volume/intensity. Conversely, OVT can last for weeks or months and might even lead to the interruption of an athlete's carrer (*Ackel-D'Elia et al., 2010*).

At least in Brazil, many physical education professionals work in gyms and exercise facilities and centers (*Da Silva, Santos & Araújo, 2016*). According to Brazilian legislation (*Brazil, 1998*), the physical education professionals are responsible for exercise prescription in these workplaces. It is reasonable to assume that these professionals present a high level of physical exertion, constituted by physical exercise performed in working environment (e.g., during group-based activities) added to their own training routine. In addition, these professionals may be exposed to other external factors to the training, such as family, economic, social and emotional problems, which may further predispose to symptoms of OVR and/or OVT syndrome (*Hjälm et al., 2007*; *Fletcher & Scott, 2010*; *Mazerolle et al., 2011*).

Thus, it is possible that this specific population has predisposing factors, signs and symptoms of OVR and OVT due to a high load and volume of exercise followed by suboptimal recovery time. Therefore, this study aimed to identify the predisposing factors, signs and symptoms of OVR and OVT in physical education professionals. We hypothesized that physical education professionals who work in gyms/exercise facilities have a high load and volume of exercise accompanied by inappropriate recovery. This might be especially true for professionals who participate in aerobic activities, which could

**Table 1  Characteristics of the professional groups.**

| | RG (n = 74) | AG (n = 20) | RAG (n = 38) | Total (n = 132) |
|---|---|---|---|---|
| Age (years) | 27.0 ± 6.0 | 27.2 ± 6.2 | 27.3 ± 6.1 | 27.1 ± 6.0 |
| Body mass (kg) | 73.3 ± 13.7 | 74.2 ± 14.1 | 73.8 ± 14.1 | 73.6 ± 13.8 |
| Height (cm) | 171.5 ± 8.9 | 171.1 ± 9.2 | 171.4 ± 9.1 | 171.4 ± 9.1 |
| Body mass index (m/kg$^2$) | 24.5 ± 3.9 | 24.9 ± 3.9 | 24.7 ± 3.9 | 24.6 ± 3.7 |
| Training session duration (hours) | 1.0 ± 0.0[a] | 2.3 ± 2.0 | 1.3 ± 0.9[a] | 1.3 ± 1.0 |
| Weekly training frequency (days) | 5.0 ± 0.9[a] | 5.6 ± 0.9 | 5.0 ± 1 | 5.1 ± 0.9 |
| Training experience (months) | 57.7 ± 71.8 | 70.5 ± 68.7 | 36.4 ± 43.3 | 53.5 ± 65.0 |
| Hazards of physical training | 28.4% | 60.0% | 42.1% | 37.1% |
| Lasting more than 15 days | 12.2% | 25.0% | 23.7% | 16.7% |

Notes.
[a] Statistically significant difference from AG ($p < 0.05$, Kruskal–Wallis test followed by test of Dunn). Values are presented as mean ± standard deviation.
RG, resistance group; AG, aerobic group; RAG, resistance and aerobic group.

potentiate OVT syndrome. Thus, these professionals could present predisposing factors, signs and symptoms of OVR and OVT.

# MATERIALS AND METHODS

## Sample

A power analysis based on a medium effect size ($d = 0.30$) for the interaction between groups > of 0.05 indicated that a sample size of 126 participants provides a statistical power of 0.85 (*Faul et al., 2009*). Based on this, the present study involved a total of 132 volunteers (83 men and 49 women). The professionals were recruited from 13 gyms located in Goiânia (Brazil) and enrolled in the study based on the following criteria: (i) performing at least one daily exercise session not shorter than 60 min, with a minimum frequency of 4 days/week (total of 4 hours/week); (ii) aged between 18 and 35 years; and (iii) all kinds of physical exercise were acceptable (*Ackel-D'Elia et al., 2010*). The selected volunteers were classified into three groups: resistance training group (RG) ($n = 74$; 50 men and 24 women), composed of physical education professionals who work and practice only resistance training (this group served as control); aerobic training group (AG) ($n = 20$; 10 men and 10 women), composed of physical education professionals who work and practice only aerobic training (running, fitness classes and others); and resistance and aerobic training group (RAG) ($n = 38$; 23 men and 15 women), composed of physical education professionals who work and practice both resistance and aerobic training (Table 1).

## Experimental design and variables analyzed

A cross-sectional study was conducted with the application of two questionnaires. The first questionnaire, adapted from *Ackel-D'Elia et al. (2010)*, aims to identify predisposing factors, signs and symptoms of OVR and OVT, and the other questionnaire, the Profile of Mood State Questionnaire (POMS), aims to identify mood disorders in the volunteers (*McNair, Lorr & Droppleman, 1971*).
Briefly, the first questionnaire aims to identify predisposing factors, signs and symptoms of OVR and OVT. It consists of 42 questions related to the work activity and/or study, obligations and external problems to training, type of physical activity practiced, motivation for class/training, patterns of rest and recovery, nutritional factors, presence of chronic diseases, medication use, monotony, fatigue and motivation, self-evaluation of training/class, performance and recovery, sleep quality, injury incidence, appetite changes, weight loss, frequency of upper respiratory tract infections and excessive sweating, volume of training/class and time of physical activity practiced. The questions were classified into 10 groups of questions related to either predisposing factors (question groups 1 to 6) or signs and symptoms (question groups 7 to 10).

The scores of the question groups were calculated as performed by *Ackel-D'Elia et al. (2010)*:

*Each volunteer was asked to make a self-assessment for each question using a five-point scale. The scale ranged from 1 (absence) to 5 (severe). Mean score (MS) for each question was calculated according to the expression: $MS = \sum (f x\ S)_n$, where f stands for the ratio between the frequency of the response for the number of alternatives and the total frequency (%) obtained for each alternative within a question, S stands for the score attributed to each alternative within a question, and n stands for the number of alternatives for each question. The final MS for a specific question group was calculated as the mean of each individual MS obtained for the selected questions fitted to a specific question group. [...] The individual MS for the three features was computed using the same formula as described above, and the final group MS was calculated as the mean of the three individual mean scores.*

The volunteer was considered with OVR/OVT when scoring higher than 4 in at least six of the 10 question groups and presented an altered mood state profile. Table 2 shows the group of questions used in the identification of predisposing factors, signs and symptoms of OVR and OVT. This questionnaire was designed based on a broad literature review about OVT and OVR and its design and validation was based on the guidelines outlined by *Juniper, Guyatt & Jaeschke (1995)*.

The self-evaluation of training, performance and recovery were evaluated by means of questions 9 (*How do you assess your exercise session?*), 10 (*How do you evaluate your physical performance?*), and 11 (*How do you evaluate your physical recovery after training sessions?*), respectively. For this, we used a modified Borg scale. For question 9, the possible answers were 'very, very light', 'very light', 'slightly light', 'a little heavy', 'heavy', 'too heavy', and 'very, very heavy'. For question 10, the possible answers were 'very, very bad', 'very bad', 'poor', 'average', 'good', 'very good', and 'very, very good'. For question 11, the possible answers were 'very, very poor', 'poor', 'reasonable', 'good', 'very good', and very, very good'. This strategy was previously used by other studies to assess breathlessness (for review, see *Bausewein et al., 2007*).

For the purposes of this study, high-intensity exercise was when the participant answered 'heavy'; 'too heavy', or 'very, very heavy' in question 9. Participants were considered to have a decrease in performance if: (1) they were located at the far right of the frequency
**Table 2** Groups of questions used to assess the OVR and OVT predisposing factors, signs and symptoms.

| Question group | | Questions |
|---|---|---|
| **Predisposing factors** | | |
| Group 1 | Work activity and/or study | 1–5 |
| Group 2 | Non-training related problems | 30 and 31 |
| Group 3 | Training motivation | 18–20 |
| Group 4 | Recovery pattern | 12–15 |
| Group 5 | Nutritional pattern | 21–25 |
| Group 6 | Hydration pattern | 26–29 |
| **Signs and symptoms** | | |
| Group 7 | Sleep patterns | 33–37 |
| Group 8 | Previous injuries | 32 |
| Group 9 | Appetite pattern changes, weight loss, superior respiratory tract infections, and excessive sweating | 38–42 |
| Group 10 | Training monotony, motivation and tiredness | 6–8, 16 and 17 |

**Notes.**
Questions 9, 10 and 11 refer to self-evaluation of training, performance and recovery and have not been computed.

distribution in at least two of questions 9, 10, and 11; or (2) answered 'heavy', 'too heavy', or 'very, very hard' to question 9; or (3) who answered 'bad', 'very bad', or 'very, very bad' to question 10 and 'poor', 'very poor', and 'very, very poor' to question 11.

The second questionnaire used, POMS, aimed to identify mood disorders in the volunteers. The POMS consists of 65 adjectives that are related to mood, using a scale from 0 to 4, where '0—no way', 1—a little, '2—moderately', '3—enough', and '4—very much'. This instrument evaluates six categories: tension-anxiety, depression-dejection, anger-hostility, vigour-activity, fatigue-inertia, and mental confusion-bewilderment (*McNair, Lorr & Droppleman, 1971*; *Spreen & Strauss, 1998*).

The scores of the POMS were calculated by adding the negative factors (tension, depression, anger, fatigue, and confusion) and subtracting the positive factor (vigour). Additionally, it was analyzed whether the volunteer has or not a positive iceberg profile. A positive iceberg profile indicates that the volunteer does not have mood disorders and is characterized by a low tension, depression, anger, fatigue and confusion score and a high vigour score. When the scores related to stress, depression, anger, fatigue and confusion are high and the vigour score is low the result is a so-called inverted (negative) iceberg profile, featuring a state of altered mood (*McNair, Lorr & Droppleman, 1971*; *Morgan et al., 1987*). The POMS questionnaire is widely utilized because of its feasibility, reliability, and validity common use in psychometric studies (*Morgan et al., 1988*; *McNair, Lorr & Droppleman, 1971*). The internal consistency reliability for POMS scales is 0.84 or greater (*Spielberger, 1972*).

## Ethical aspects

All participants were informed of the potential risks and benefits of the study and signed an informed consent form. All experimental procedures were approved (no 2.259.846) by the

**Table 3  Predisposing factors, signs, and symptoms of OVR and OVT between professional groups.**

| | Question group (1 to 10) | No. of questions | RG | AG | RAG | Total raverage |
|---|---|---|---|---|---|---|
| **Predisposing factors** | | | | | | |
| 1 | Work activity and/or study | 5 | 3.4 | 3.5 | 3.6 | 3.5 |
| 2 | Non-training-related problems | 2 | 2.3 | 2.1 | 2.8 | 2.4 |
| 3 | Training motivation | 3 | 3.5 | 3.4 | 3.8 | 3.6 |
| 4 | Recovery pattern | 4 | 3.5 | 3.2 | 3.0 | 3.3 |
| 5 | Nutritional pattern | 5 | 2.0 | 1.9 | 1.9 | 2.0 |
| 6 | Hydration pattern | 4 | 2.1 | 2.2 | 2.1 | 1.8 |
| **Signs and symptoms** | | | | | | |
| 7 | Sleep pattern | 5 | 2.2 | 2.4 | 2.3 | 2.2 |
| 8 | Previous injuries | 1 | 2.4 | 3.8 | 3.6 | 2.9 |
| 9 | Appetite pattern changes, weight loss, SRTI and excessive sweating | 5 | 1.5 | 1.7 | 1.6 | 1.5 |
| 10 | Training monotony, motivation and tiredness | 5 | 2.4 | 2.7 | 2.6 | 2.5 |
| **Mean±SD** | | | 2.5 ± 0.7 | 2.7 ± 0.7 | 2.7 ± 0.8 | 2.6 ± 0.7 |

**Notes.**

SD, standard deviation; SRTI, superior respiratory tract infection; RG, resistance group; AG, aerobic group; RAG, resistance and aerobic group.

University Human Research Ethics Committee and conformed to the principles outlined in the Declaration of Helsinki.

## Statistical analysis

Data were entered into an Excel spreadsheet (Office 2016; Microsoft, Redmond, WA, USA) and imported into Statistical Package for the Social Science (SPSS) version 23.0 (IBM Corp., Armonk, NY, USA) for statistical analysis. Data were presented as mean ± standard deviation and absolute and relative frequencies. To test the normality of the sample, we used the Kolmogorov–Smirnov test. As no data about domains of the POMS profile followed a normal distribution, the comparison between groups (RG vs. AG vs. RAG) was done using the Kruskal–Wallis test followed by Dunn's pos $t$-test. Fisher's exact test was applied to evaluate the association between the presence of OVT and group. A significance level of 5% was set for all statistical procedures.

## RESULTS

The results showed that none of the question groups for any of the analyzed groups presented scores above four (Table 3).

As mentioned in the methods, a decreased performance was considered in those volunteers who were located at the extreme right in at least two of the following three self-evaluated aspects: training (Fig. 1A); performance (Fig. 1B); and recovery (Fig. 1C). In this context, only 2.7%, 15% and 18.4% of the RG ($n = 2$), AG ($n = 3$), and RAG ($n = 7$), respectively, were considered to have decreased performance. The exact Fisher's test showed that distribution frequency was significantly different between groups ($p = 0.009$).

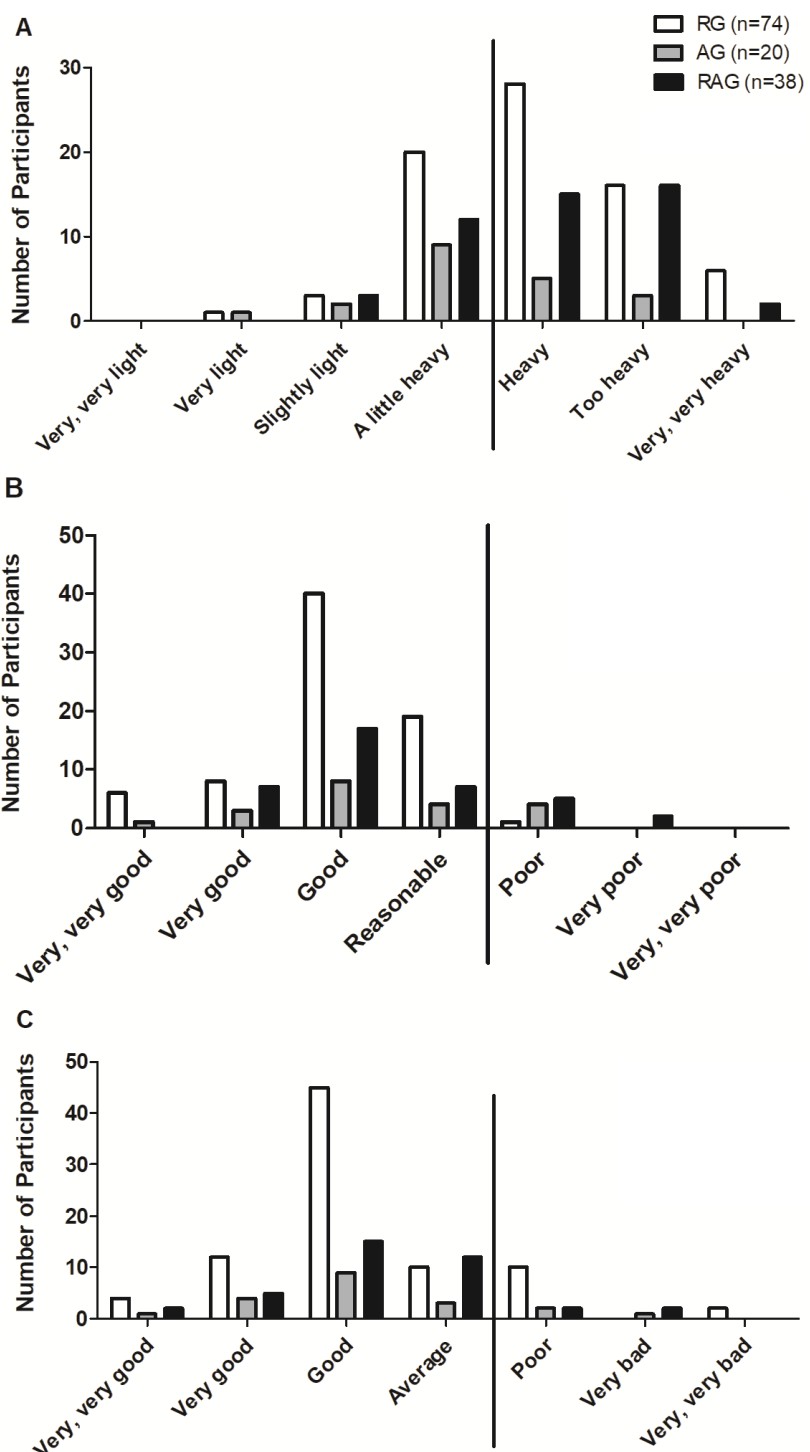

**Figure 1 Self-evaluation of the training (A), performance (B), and recovery (C) of professionals according to the groups.** RG, resistance training group; AG, aerobic training group; RAG, resistance and aerobic training group.

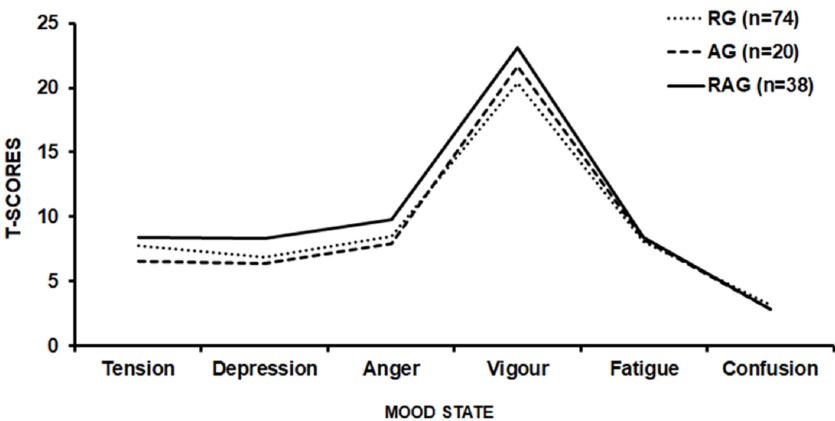

**Figure 2 Graphic POMS profile of the resistance training group (RG), aerobic training group (AG), and resistance and aerobic training group (RAG).**

Regarding the assessment of mood state by POMS, 67.5%, 80.0% and 60.5% of the RG ($n = 50$), AG ($n = 16$), and RAG ($n = 23$), respectively, were classified as absent of mood disorders (*Standard Graphic Iceberg*—Fig. 2). There were no statistical differences ($p = 0.82$) in the total mood disorder among RG ($13.9 \pm 24.5$), AG ($10.3 \pm 25.1$) and RAG ($14.6 \pm 27.9$) groups.

## DISCUSSION

The primary aim of this study was to evaluate the risk of OVR and OVT (related to physical/mental work stress) in physical education professionals. The results showed that the sample did not present predisposing factors, signs and symptoms of OVR and OVT. In addition, we observed a low presence of performance impairment. However, it is noteworthy that workplaces—especially for professionals whose bodies are a working tool—can place workers at a higher risk of injuries, disabilities, morbidities, and mortality (*Knoblauch & Cassaro, 2017*).

Our results are in line with *Ackel-D'Elia et al. (2010)*, who assessed the prevalence of predisposing factors, signs and symptoms of OVR and OVT in 413 members of 17 gyms in São Paulo (Brazil). The authors found a low prevalence of signs and symptoms of OVR and OVT, including among individuals classified as belong to a higher risk subgroup.

The findings of the present study do not support the initial hypothesis, that the high weekly volume of exercise (that includes own training session and exercise performed professionally), would put physical education professionals at a high risk of developing OVR and/or OVT syndrome. In fact, when we compare the weekly amount of exercise performed by volunteers evaluated ($5.1 \pm 0.9$ h) with that recommended by scientific societies, such as the American College of Sports Medicine (ACSM) (*Garber et al., 2011*), we found that the weekly volume of exercise is greater than recommended for the maintenance and development of cardiorespiratory and neuromuscular fitness (*Garber et al., 2011*). The ACSM recommends that individuals should engage in a moderate-intensity program for at least 30 min a day for five or more days a week, totaling a minimum of 150 min (2.5 h)

per week or perform an intense training for at least 20 min a day for three or more days a week, totalizing a minimum of 75 min (1.25 h) per week (*Garber et al., 2011*).

Considering the weekly volume recommended by the ACSM for moderate exercise (150 min) (*Garber et al., 2011*), the assessed volunteers trained more than twice the recommended time/volume (∼400 min). However, when the weekly training volume of the sample is compared with that of high-level athletes, 25–38 hours/week (*Woods et al., 2017*), it might be considered low. In this regard, the findings of the present study appear consistent; that is, they show an absence of predisposing factors, signs and symptoms of OVR and OVT in physical education professionals. Even if we evaluate only the AG, the average weekly exercise of the participants in this group is around 13 hours/week, also very distant from that found for professional athletes.

Another variable related to the development of OVR and OVT is training intensity (*Purvis, Gonsalves & Deuster, 2010*). In the present study, with regard to training intensity (question 14), 80 (60.6%) of the 132 subjects reported that their training intensity was at least as a little heavy. Many authors advocate that high-intensity exercise is necessary to evoke physiological adaptations, and consequently to improve physical performance (*Rogero, Mendes & Tirapegui, 2005*; *Bosquet et al., 2007*). However, coaches should be aware of training load and fatigue-related parameters in order to avoid OVR symptoms. In this context, it is clear that training stimulus must be accompanied by adequate periods of recovery and a proper nutrition and hydration (*Rogero, Mendes & Tirapegui, 2005*; *Bosquet et al., 2007*).

Regarding recovery (question 16), 120 (90.9%) reported a 'reasonable', 'good', 'very good' or 'very, very good' recovery. Therefore, even though the training intensity was 'a little heavy' for more than half of the volunteers, they reported an adequate recovery. As a result of a fairly heavy training and a considerable recovery, 91.6% ($n = 121$) of the volunteers achieved at least regular performance.

Sleep disorders, occupational hazards, injuries and illnesses could impact professional and worker overall health (*Kalliny & McKenzie, 2017*). Sportsmen and women with OVR and OVT usually reported sleep disorders (*Winsley & Matos, 2010*). In this regard, the participants of the present study can be considered as having no sleep disorders, with the score of three groups showing an average below 3.0. A similar pattern was found for groups of questions that assessed external obligations to training, nutritional factors, hydration pattern, changes in appetite, weight loss, respiratory tract infections, excessive sweating, monotony, fatigue and motivation in relation to training. Particularly, in relation to nutritional factors, it has been suggested that, to ensure a more efficient replacement of energy stocks, especially muscle glycogen, the post-workout meal should be eaten in the first hours of post-exercise recovery (*Beelen et al., 2010*). Of the 132 volunteers evaluated, 90.9% ($n = 120$) ate within two hours after the training session.

Regarding monotony, we found that RG, AG and RAG showed low tiredness and monotony in relation to training (scores lower than 3.0). This suggests that monotony was not singled out as a factor that interferes in the training sessions for the volunteers. This data combined with the results of training intensity, recovery and performance suggest that the training carried out by volunteers was adequate. Considering that the volunteers

are physical education professionals, it is reasonable to assume that they had current and scientifically relevant information about exercise prescription and training planning.

Finally, regarding mood, 64.4% ($n = 89$) volunteers had an absence of signs and symptoms of mood disorders (positive iceberg), while 32.6% ($n = 43$) had an outside graphic pattern or inverted/negative iceberg. Despite the relatively high number of individuals with altered mood state, it seems prudent to not associate this fact to the characteristics of the training, but possibly to psychosocial stressors related to work/modern life, since the RG, AG and RAG had an average score close to 4.0 in the group of questions related to the work activity and/or study (one of the highest scores among those evaluated).

### Study limitations

Firstly, the volunteers did not undergo biochemical (creatine kinase, testosterone, and cortisol) and physiological (heart rate variability and oxygen uptake) analysis. Such analysis could provide valuable information about the physiological status of volunteers, since several studies have demonstrated that these biochemical and physiological measures are changed in sportsmen and women with OVT (*Halson et al., 2003*; *Silva, Santhiago & Gobatto, 2006*; *Cadegiani & Kater, 2017a*; *Cadegiani & Kater, 2017b*). Secondly, the volunteers were not submitted to objective tests to evaluate their performance, such as a cardiorespiratory exercise test. Thirdly, the present study cannot differentiate the states of OVR and OVT. Fourthly, as for all studies employing questionnaires, the present results rely on the honesty and level of recall of respondents. Nevertheless, we believe that these limitations do not prevent the study's conclusions from being drawn.

## CONCLUSION

Despite the relatively high-volume work/training performed by the physical education professionals surveyed, we can conclude that they did not have predisposing factors, signs and symptoms of OVR and OVT. Additionally, no differences between RG, AG and RAG were identified.

## ACKNOWLEDGEMENTS

We would like to thank the participants for their effort and commitment to the research project.

### Funding
The authors received no funding for this work.

### Competing Interests
The authors declare there are no competing interests.

## Author Contributions

- Ricardo B. Viana conceived and designed the experiments, performed the experiments, analyzed the data, prepared figures and/or tables, authored or reviewed drafts of the paper, approved the final draft.
- Paulo Gentil, Carlos A. Vieira, Mário H. Campos, Douglas A.T. Santos and Wellington F. Silva conceived and designed the experiments, analyzed the data, authored or reviewed drafts of the paper, approved the final draft.
- Vinício S. Lorenço conceived and designed the experiments, performed the experiments, analyzed the data, authored or reviewed drafts of the paper, approved the final draft.
- Marilia S. Andrade and Rodrigo L. Vancini conceived and designed the experiments, analyzed the data, contributed reagents/materials/analysis tools, authored or reviewed drafts of the paper, approved the final draft.
- Claudio A.B. de Lira conceived and designed the experiments, performed the experiments, analyzed the data, contributed reagents/materials/analysis tools, prepared figures and/or tables, authored or reviewed drafts of the paper, approved the final draft.

## Human Ethics

The following information was supplied relating to ethical approvals (i.e., approving body and any reference numbers):

The Federal University of Goiás granted Ethical approval to carry out the study within its facilities (Ethical Application Ref: 2.259.846).

## Data Availability

The raw data was uploaded as a Supplemental File.

## Supplemental Information

Supplemental information for this article can be found online at http://dx.doi.org/10.7717/peerj.4994#supplemental-information.

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
