# Peer review of "Identifying the predisposing factors, signs and symptoms of overreaching and overtraining in physical education professionals"

_PeerJ, doi:10.7717/peerj.4994_

## Round 0.1 · original submission · Major Revisions

Dear authors:

Your manuscript was evaluated by two expert reviewers. Please address the reviewer´s concerns in detail.

Reviewer 1 ·

Basic reporting

I commend the authors for their work. The manuscript is clearly written in professional, unambiguous language. The paper is easily-read and the findings have important implications for this population. The theoretical background provided is enough and article structure is professional and clear for readers.

However, there are some concerns. Mainly, related to the sample size (and this might affect to the potential and impact of this paper). Additionally, some minor corrections have been requested.

Experimental design

Some comments are described below:

*Abstract:
- The reviewer is not a native english speaker, but some mistakes have been detected. Please, check the entire manuscript according to this.
- These abreviations must be explained here (and not in the next sentence)

*Introduction:
- I really enjoyed reading this intro section. It is well organised and easy for readers to undersntand the importance of this study.
- Lines 69-76: Any previous study to support this?

*Material and methods:
-Sample: I am not convinced about the convenience of this sample size. The paper would be reinforced if the authors provide the power of this sample size. This population (sports and exercise professionals) is huge, and a N=132 is not close to an infinite sample size...
- Lines 91-92: Many papers have been previously published about OVT or OVR in athletes. All those papers report training loads much higher than 4 h/wk. Take a look, for example, to this paper in which the author describe the training plan of a professional triathlete:
https://journals.humankinetics.com/doi/abs/10.1123/ijspp.2013-0245
Based on that, the criterion used for the authors to talk about OVR or OVT seems not to be enough... Could you report any cases of OVT or OVR with similar training loads?
- experimental design: It is quite controversial to talk about overtraining from a cross-sectional design...even more, if physiological (HRV, HR...) nor biochemical parameteres (cortisol, testosterone...) were nor monitored or measured.
- Questionnaires: Please, as well as the validation paper, provide the psychometric properties of both questionnaires

Validity of the findings

*Conclusions: lines 293-294
Please, add a reference to support this sentence (scientific consensus)

Additional comments

I commend the authors for their work. The manuscript is clearly written in professional, unambiguous language. The paper is easily-read and the findings have important implications for this population. The theoretical background provided is enough and article structure is professional and clear for readers.

However, there are some concerns. Mainly, related to the sample size (and this might affect to the potential and impact of this paper). Additionally, some minor corrections have been requested.

Reviewer 2 ·

Basic reporting

Title.
- In regards to the title, it is not clear what is meant by "sports and exercise professionals". It can be understand in many who for example: athletes or any professional dedicated to working with athletes or those who exercise (sports psychologist, sports nutritionist, etc.). I suggest that the title be more specific as to those who this research is about: personal trainers, coaches or other. I am suggesting that all wording written as: "sports and exercise professionals / workers", be clarified so as to understand,without a doubt, who you are talking about.

English language.
-The English language should be improved to ensure that an international audience can clearly understand your text. Some examples where the language could be improved include line 100 : practice (no practise).

Abstract.
- Line 24 does not describe the abbreviation of OVR and OVT. These appear on lines 26 but should be on line 24, which is where these two abbreviations first appear.

Introduction.
- Line 78: I suggest: ... symptoms of OVR and/or OVT ...

Discussion.
- Line 237-238-239, talks about shock microcycle training (high intensity) that improved the performance of athletes (tennis players). I believe they do not have much relationship with sports professionals (trainers). I suggest that you can include some study that is related to the coaches.

introduction, discussion.
On many occasions it is mentioned that personal trainers/coaches intentionally modified the athletes (volume / intensity) load training but then talk about sports and exercise professionals (coaches or personal trainers) are exposed to symptoms of overtraining due to their own personal trainings, in addition to possibly be experiencing personal problems. This is not clear, because first it refers to athletes and then to coaches.

Experimental design

The Borg scale has been modified. I suggest including some reference regarding the validity of this modified scale. For example, the Borg CR100 scale has been used as a complement to CR10 in some studies.

Validity of the findings

Conclusions are well stated, linked to original research question & limited to supporting results.

Additional comments

- It is important to know/ask how many years the personal trainer/coach has been performing the work routine? Have you considered this question in your survey?

---

## Round 0.2 · accepted · Accept

I am pleased to inform you of the official acceptance of your manuscript for publication in PeerJ.

Thank you very much for the opportunity to review your manuscript and congratulations.

Reviewer 1 ·

Basic reporting

Thanks to the authors for taking my comments and suggestions into consideration. I really think the paper has substantially improved.
The writing is much clearer and definitively easier for readers to follow and understand the text; and the changes done in the methods section (sample size and new references for the questionnaires used) along with the information added to the discussion section, improved the quality of the manuscript.
Congratulations on your work!

Experimental design

No comment

Validity of the findings

No comment

Additional comments

Thanks to the authors for taking my comments and suggestions into consideration. I really think the paper has substantially improved.
The writing is much clearer and definitively easier for readers to follow and understand the text; and the changes done in the methods section (sample size and new references for the questionnaires used) along with the information added to the discussion section, improved the quality of the manuscript.
Congratulations on your work!

Reviewer 2 ·

Basic reporting

Thank you. All suggested changes have been modified.

Experimental design

thanks for including the reference.

Validity of the findings

Thank you for including these questions in the table 1.

Additional comments

Good job with this new version.